# How informative is the Approximation Error from Tensor Decomposition for Neural Network Compression?

**Jetze Schuurmans**[1,2], **Kim Batselier**[2], **Julian F. P. Kooij**[1]
[1] Cognitive Robotics Department (CoR), [2] Delft Center for Systems and Control (DCSC)
3mE Faculty, TU Delft, The Netherlands
{j.t.schuurmans,k.batselier,j.f.p.kooij}@tudelft.nl

## Abstract

Tensor decompositions have been successfully applied to compress neural networks. The compression algorithms using tensor decompositions commonly minimize the approximation error on the weights. Recent work assumes the approximation error on the weights is a proxy for the performance of the model to compress multiple layers and fine-tune the compressed model. Surprisingly, little research has systematically evaluated which approximation errors can be used to make choices regarding the layer, tensor decomposition method, and level of compression. To close this gap, we perform an experimental study to test if this assumption holds across different layers and types of decompositions, and what the effect of fine-tuning is. We include the approximation error on the features resulting from a compressed layer in our analysis to test if this provides a better proxy, as it explicitly takes the data into account. We find the approximation error on the weights has a positive correlation with the performance error, before as well as after fine-tuning. Basing the approximation error on the features does not improve the correlation significantly. While scaling the approximation error commonly is used to account for the different sizes of layers, the average correlation across layers is smaller than across all choices (i.e. layers, decompositions, and level of compression) before fine-tuning. When calculating the correlation across the different decompositions, the average rank correlation is larger than across all choices. This means multiple decompositions can be considered for compression and the approximation error can be used to choose between them.

## 1 Introduction

Tensor Decompositions (TD) have shown potential for compressing pre-trained models, such as convolutional neural networks, by replacing the optimized weight tensor with a low-rank multi-linear approximation with fewer parameters (Jaderberg et al., 2014; Lebedev et al., 2015; Kim et al., 2016; Garipov et al., 2016; Kossaifi et al., 2019a). Common compression procedures (Lebedev et al., 2015; Garipov et al., 2016; Hawkins et al., 2021) work by iteratively applying TD on a selected weight tensor, where each time several *decomposition choices* have to be made regarding (i) the layer to compress, (ii) the type of decomposition, and (iii) the compression level. Selecting the best hyperparameters for these choices at a given iteration requires costly re-evaluating the full model for each option. Recently, Liebenwein et al. (2021) suggested comparing the *approximation errors* on the decomposed weights as a more efficient alternative, though they only considered matrix decompositions for which analytical bounds on the resulting performance exist. These bounds rely on the Eckhart-Young-Mirsky theorem. For TD, no equivalent theorem is possible (Vannieuwenhoven et al., 2014). While theoretical bounds are not available for more general TD methods, the same concept could still be practical when considering TDs too. We summarize this as the following general assumption:

**Assumption 1.** *A lower TD approximation error on a model's weight tensor indicates better overall model performance after compression.*

While this assumption appears intuitive and reasonable, we observe several gaps in the existing literature: First, most existing TD compression literature only focuses on a few decomposition choices, e.g. fixing the TD method (Lebedev et al., 2015; Kim et al., 2016). Although various error measures and decomposition choices have been studied in separation, no prior work systematically compares different decomposition errors across multiple decomposition choices. Second, different decomposition errors with different properties have been used throughout the literature (Jaderberg et al., 2014), and it is unclear if some error measure should be preferred. Third, a benefit of TD is that no training data is needed for compression, though if labeled data is available, more recent methods combine TD with a subsequent fine-tuning step. Is the approximation error equally valid for the model performance with and without fine-tuning?

Overall, to the best of the authors' knowledge, no prior work investigates if and which decomposition choices for TD network compression can be made using specific approximation errors. This paper studies empirically to what extent a single decomposition error correlates with the compressed model's performance across varied decomposition choices, identifying how existing procedures could be improved, and providing support for specific practices. Our contributions are as follows:

- A first empirical study is proposed on the correlation between the approximation error on the model weights that result from compression with TD, and the performance of the compressed model[1]. Studied decomposition choices include the layer, multiple decomposition methods (CP, Tucker, and Tensor Train), and level of compression. Measurements are made using several models and datasets. We show that the error is indicative of model performance, even when comparing multiple TD methods, though useful correlation only occurs at the higher compression levels.

- Different formulations for the approximation error measure are compared, including measuring the error on the features as motivated by the work Jaderberg et al. (2014); Denil et al. (2014) which considers the data distribution. We further study how using training labeled data for additional fine-tuning affects the correlation.

## 2 RELATED WORK

There is currently no systematic study on how well the approximation error relates to a compressed neural network's performance across multiple choices of network layers, TD methods, and compression levels. We here review the most similar and related studies where we distinguish works with theoretical versus empirical validation, different approximation error measures, and the role of fine-tuning after compression.

The relationship between the approximation error on the weights and the performance of the model was studied by theoretical analysis for matrix decompositions. Liebenwein et al. (2021) derive bounds on the model performance for SVD-based compression on the convolutional layers, and thus motivate that the SVD approximation error is a good proxy for the compressed model performance. Arora et al. (2018) derive bounds on the generalization error for convolutional layers based on a compression error from their matrix projection algorithm. Baykal et al. (2019) show how the amount of sparsity introduced in a model's layers relates to its generalization performance. While these works show that some theoretical bounds can be found for specific compression methods, such bounds are not available for TDs in general. Other works, therefore, study the relationship for TD empirically. For instance, Lebedev et al. (2015) show how CP decomposition rank affects the approximation error, and the resulting accuracy drop as the rank is decreased. Hawkins et al. (2022) observe that, for networks with repeated layer blocks, the approximation error depends on the convolutional layers within the block.

When considering the model's final task performance, the approximation error on the weights might not be the most relevant measure. To consider the effect on the actual data distribution, Jaderberg et al. (2014) instead propose to compute an error on the approximated output features of a layer after its weights have been compressed. They found that compressing weights by minimizing the error on features, rather than the error on the weights, results in a smaller loss in classification accuracy. However, Jaderberg et al. (2014) do not fine-tune the decomposed model, and only use a toy model with few layers. Denil et al. (2014) try to capture the information from the data via the empirical

---

[1]The code for our experiments is available at `https://github.com/JSchuurmans/tddl`.

covariance matrix. Although this method eliminates the need for multiple passes over the data during the compression step, it is limited to a two-dimensional case. Eo et al. (2021) forego looking at a compression error altogether, and selects the rank based on the accuracy on the validation set. This requires the labels to be present and a forward pass through the whole network, even if the compressed layer is near the input.

Several norms have been used in the literature to quantify the approximation error, which is the difference between the pretrained weights and the compressed weights. The works that decompose pretrained layers (Denton et al., 2014; Jaderberg et al., 2014; Lebedev et al., 2015; Novikov et al., 2015; Kim et al., 2016) explicitly minimize the Frobenius norm. Hawkins et al. (2021); Liebenwein et al. (2021) calculate the relative Frobenius norm, i.e., the norm of the error proportional to the norm of the pretrained weights, to compare the error for layers of different sizes. Still, it remains unclear which error measure is most informative for the compressed model's final performance.

In practice, when training data is still available for compression, fine-tuning for the target task after compressing weights could recover some of the lost performance (Denton et al., 2014; Lebedev et al., 2015; Kim et al., 2016). Adding fine-tuning results in a three-step process: pretrain, compress and fine-tune. Optimization thus alternates between minimizing the error of respectively the features, the weights, and finally the features again. While Denton et al. (2014); Kim et al. (2016) compare compressed model performance before and after fine-tuning, they do not investigate how the fine-tuned network performance relates to the weight compression error. Lebedev et al. (2015) does study the compression error for CP decomposition, but only reports performance with fine-tuning.

## 3   METHODOLOGY

We consider the task of compressing a pretrained neural network with TD. While TD is a general technique that could be applied to many types of layers, the focus will be on convolutional layers. Due to their ubiquity and suitability to compare different types of higher-order decompositions, as the layer weights are four-dimensional tensors.

Generally, a compression procedure will iteratively apply TD to the weights of selected layers, making several choices on how and what weight tensor to decompose, while ideally maintaining as much of the original network's performance. In its original uncompressed form, the full-rank weight tensors $\mathbf{W} \in \mathbb{R}^{C \times H \times W \times T}$ of a layer represent a local optimum in the network's parameter space with respect to the training data and loss, where $C$ is the number of input channels, $H$ and $W$ are the height and width of the convolutional kernel, and $T$ is the number of output channels. When a TD is applied to the weights of a specific layer, this results in a factorized structure $\widetilde{\mathbf{W}}$ composed of multiple smaller tensor multiplications, which replaces the original weights in the network. Each time TD is applied, several decomposition choices need to be evaluated:

1. *Layer:* The layer $l$ from the set of network layers $\mathbb{L} = \{1, 2, \cdots, L\}$ to decompose.
2. *Method:* The type of TD method $m \in \mathbb{M} = \{\text{CP}, \text{Tucker}, \text{Tensor Train}\}$. The decomposition determines the factorized structure of $\widetilde{\mathbf{W}}$.
3. *Compression:* The compression level $c \in \mathbb{C}$ for the selected layer. Here $\mathbb{C} \subset (0, 1]$ is some finite set of testable levels, and $c = 0.75$ means the number of parameters is reduced by 75% and the factorized layer contains only 25% of the parameters. A given compression level is achieved by decomposing the tensor to some rank $\mathcal{R}$, depending on the selected TD method (see Section 3.3).

We will refer to $\mathbb{H} = \mathbb{L} \times \mathbb{M} \times \mathbb{C}$ as the set with possible hyperparameter values for $(l, m, c)$ to consider. Note that compression procedures in the literature might only consider a subset of these choices. For example, a procedure might fix the layer for a given iteration or only consider a single TD method. In practice, it is computationally infeasible to evaluate the compressed network's performance for every possible hyperparameter choice at every compression iteration, especially when optimizing for performance after fine-tuning. Instead, automated compression procedures will efficiently compare an approximation error $a_i = e(\widetilde{\mathbf{W}}_i, \mathbf{W})$ between the original and decomposed weights using a particular choice of hyperparameters $h_i \in \mathbb{H}$. In doing so one relies on Assumption 1 that a lower approximation error is indicative of better compressed performance $p_i$. If annotated training data is available, additional network fine-tuning on the decomposed structure could result in improved performance $p_i^\star > p_i$.

In this work, we propose to focus on a single iteration and investigate Assumption 1 in isolation of any specific compression procedure. Our aim is thus to assess how well computing an approximation error $e$ can predict the optimal compression choice from some hypothesis set $\mathbb{H}$, i.e. the choice that results in the lowest compressed network performance error $p$, or even performance after fine-tuning $p^\star$. We will study the correlation between approximation error and model performance empirically in our experiments, using the procedure and correlation metric explained in Section 3.1. Details on the different approximation errors that we will explore are covered in Section 3.2. Finally, the considered TD methods are explained in Section 3.3.

### 3.1 EMPIRICAL EVALUATION OF ERROR-PERFORMANCE CORRELATION

Our proposed empirical evaluation procedure will evaluate a large set of hyperparameters $\mathbb{H} = \{h_1, h_2, \cdots\}$ on multiple convolutional neural networks and datasets (see Section 4) for different options of approximation error metric (see Section 3.2). For a given model, dataset, and approximation error metric $e$, the procedure evaluates for each set of hyperparameter choices $h_i \in \mathbb{H}$ the approximation error $a_i = e(\widetilde{\mathbf{W}}_i, \mathbf{W})$, the model performance error $p_i$ on the validation split, and the model performance error $p_i^\star$ after additional fine-tuning on the training data. We thus obtain sets of measurements $\mathbb{A} = \{a_1, a_2, \cdots\}$, $\mathbb{P} = \{p_1, p_2, \cdots\}$ and $\mathbb{P}^\star = \{p_1^\star, p_2^\star, \cdots\}$ for $\mathbb{H}$.

When comparing two sets of hyperparameters $h_i \in \mathbb{H}$ and $h_j \in \mathbb{H}$, we want to establish if the set with the smaller approximation error results in a smaller performance error of the model. In other words, the concordance of pairs of measurements needs to be established. Concordant pairs have a larger (smaller) performance error when the approximation error is larger (smaller) between two sets of hyperparameter choices, i.e. $i$ and $j$ are concordant if $a_i > a_j$ and $p_i > p_j$ or if $a_i < a_j$ and $p_i < p_j$, and discordant otherwise. Kendall's $\tau$ is a measure for the rank correlation (Kendall, 1938) or ordinal association between two order sets, in our case between approximation errors $e$ and a model performances $\mathbb{P}$ (or $\mathbb{P}^\star$). To avoid confusion with the concept of tensor rank, we will refer to Kendall's $\tau$ simply as *correlation*. For this correlation measure, the difference between the number of concordant pairs ($k$) and discordant pairs ($d$) is scaled with the binomial coefficient $m(m-1)/2$ to account for the different ways two measurements can be sampled from a total of $m$ measurements:

$$\tau = 2(k - d)/(m(m - 1)). \tag{1}$$

Kendall's $\tau$ can be interpreted as follows: $\tau = 1$ indicates a perfect positive rank correlation, $\tau = 0$ no correlation, and $\tau = -1$ a strong negative correlation. For a set of hyperparameters $\mathbb{H}$, a useful approximation error $e$ would thus result in a $\tau$ close to $\pm 1$, indicating it is predictive of the model's performance. Note that Kendall's $\tau$ does not depend on assumptions about the underlying distribution, whereas Pearson correlation assumes a linear relationship between the two measurements. Kendall's $\tau$ is used over Spearman's $\rho$ because the interpretation of con- and discordance pairs for Kendall's $\tau$ is closely related to our use case of choosing between hyperparameter sets.

### 3.2 APPROXIMATION ERRORS

We now discuss various measures $e(\widetilde{\mathbf{W}}, \mathbf{W})$ to quantify the approximation error. The basis is to compute some norm on the difference between these tensors, in this work we use the Frobenius norm as is common in the literature (Lebedev et al., 2015; Hawkins et al., 2022). We shall consider three options to scale the norm, which could help make the error more robust when comparing hypotheses with different layers. Additionally, we can consider two options to compute the error on, namely either directly the weights or on the features. In total, we shall thus explore six different approximation errors in this work. An overview is presented in Table 1.

**Normalization**   The norm between the difference of the weights is referred to as **absolute** norm and is used in the objective function when decomposing the pretrained weights. The **relative** norm is used in TD literature to compare errors between different layers (Lebedev et al., 2015; Hawkins et al., 2022), as it is invariant to the size of the weights. Alternatively, the norm of the difference can be **scaled** to account for the number of parameters, while keeping the distance from the weights.

**Target tensor**   The most common option is to compute approximation error on the decomposed layer's *weights*, $\mathbf{W}$. However, Jaderberg et al. (2014) achieved promising results basing the decomposition on the approximation error of the features. Errors in some elements of the weight tensor

might be more permissible if they do not affect the resulting feature space. We therefore also consider the expected error on the *features* $\mathbf{F} = \mathbf{X} \cdot \mathbf{W}$, which is the output tensor of the layer after convolving its input with weights $\mathbf{W}$ on input data $\mathbf{X}$. Likewise, approximated weights $\widetilde{\mathbf{W}}$ result in an approximated $\widetilde{\mathbf{F}}$. In practice, computing the feature space requires input data and is computationally more demanding than computing the weight approximation error. However, it is potentially more representative of an approximation's effect on the output, plus unlike a later fine-tuning step, it could even be used if only data without labels is available.

Table 1: Overview of the approximation errors $e(\widetilde{\mathbf{W}}, \mathbf{W})$ considered in our evaluation. Rows show the target tensor to evaluate the error on, and columns the different options for error normalization. $n_{\mathbf{W}}$ and $n_{\mathbf{F}}$ are the number of elements in the weight tensor $\mathbf{W}$ or feature tensor $\mathbf{F}$ respectively.

|  | **Absolute** | **Relative** | **Scaled** |
|---|---|---|---|
| Weight | $\|\|\mathbf{W} - \widetilde{\mathbf{W}}\|\|$ | $\|\|\mathbf{W} - \widetilde{\mathbf{W}}\|\|/\|\|\mathbf{W}\|\|$ | $\|\|\mathbf{W} - \widetilde{\mathbf{W}}\|\|/n_{\mathbf{W}}$ |
| Feature | $\mathbb{E}_{\mathbf{X}}\left[\|\|\mathbf{F} - \widetilde{\mathbf{F}}\|\|\right]$ | $\mathbb{E}_{\mathbf{X}}\left[\|\|\mathbf{F} - \widetilde{\mathbf{F}}\|\|/\|\|\mathbf{F}\|\|\right]$ | $\mathbb{E}_{\mathbf{X}}\left[\|\|\mathbf{F} - \widetilde{\mathbf{F}}\|\|/n_{\mathbf{F}}\right]$ |

### 3.3 TENSOR DECOMPOSITION METHODS

Our experiments shall consider three popular decomposition methods for convolutional layers, namely CP (Denton et al., 2014; Jaderberg et al., 2014; Lebedev et al., 2015), Tucker (Kim et al., 2016), and Tensor Train (TT) (Garipov et al., 2016). During the decomposition step the decomposed weights are found by minimizing the approximation error between the pretrained weights and the estimated decomposition: $\arg\min_{\widetilde{\mathbf{W}}} \|\|\mathbf{W} - \widetilde{\mathbf{W}}\|\|$. For CP this is done with ALS (Carroll & Chang, 1970; Harshman, 1972), for Tucker with HOSVD (De Lathauwer et al., 2000), and TT-SVD (Oseledets, 2011) for Tensor Train. The ALS algorithm requires a random initialization. We sample from a uniform distribution [0,1) using `Tensorly` (Kossaifi et al., 2019b). The desired compression level is achieved by finding the corresponding rank, using the package `Tensorly-Torch` (Kossaifi et al., 2019b). The ranks used for CP, Tucker, and TT are given in Appendix A.1. For completeness, we list all considered decompositions with a 4-way tensor $\mathbf{W}$.

**CP decomposition**    A rank-$R$ CP decomposition (Hitchcock, 1927) sums $R$ rank-one tensors:

$$\widetilde{\mathbf{W}}^{\text{CP}}_{c,y,x,t} = \sum_{r=1}^{R} \boldsymbol{C}_{c,r} \boldsymbol{Y}_{y,r} \boldsymbol{X}_{x,r} \boldsymbol{T}_{t,r}. \tag{2}$$

**Tucker decomposition**    A Tucker decomposition (Tucker, 1966) is distinct from a CP by the Tucker core $\mathbf{G} \in \mathbb{R}^{R_1 \times R_2 \times R_3 \times R_4}$. The Tucker rank is defined as the four-tuple $(R_1, R_2, R_3, R_4)$. Since the dimensions of the convolutional weights are small with respect to the width and height dimensions, it is computationally more efficient to contract these with the Tucker core $\mathbf{G}$ and form a new core $\mathbf{H} = \mathbf{G} \times_2 \boldsymbol{Y} \times_3 \boldsymbol{X}$, where $\times_n$ is the n-mode product (Appendix A.2):

$$\widetilde{\mathbf{W}}^{\text{Tucker}}_{c,y,x,t} = \sum_{r_1=1}^{R_1} \sum_{r_2=1}^{R_2} \sum_{r_3=1}^{R_3} \sum_{r_4=1}^{R_4} \mathbf{G}_{r_1,r_2,r_3,r_4} \boldsymbol{C}_{c,r_1} \boldsymbol{Y}_{y,r_2} \boldsymbol{X}_{x,r_3} \boldsymbol{T}_{t,r_4} = \sum_{r_1=1}^{R_1} \sum_{r_4=1}^{R_4} \mathbf{H}_{r_1,y,x,r_4} \boldsymbol{C}_{c,r_1} \boldsymbol{T}_{t,r_4}. \tag{3}$$

**Tensor Train decomposition**    Another alternative is the Tensor Train decomposition (Oseledets, 2011), which decomposes a given tensor as a linear chain of 2-way and 3-way tensors, where the first and last tensors are 2-way. The TT-rank in our four-dimensional case is the 3-tuple $(R_1, R_2, R_3)$:

$$\widetilde{\mathbf{W}}^{\text{TT}}_{c,y,x,t} = \sum_{r_1=1}^{R_1} \sum_{r_2=1}^{R_2} \sum_{r_3=1}^{R_3} \boldsymbol{C}_{c,r_1} \mathbf{Y}_{r_1,y,r_2} \mathbf{X}_{r_2,x,r_3} \boldsymbol{T}_{r_3,t}. \tag{4}$$

## 4    EXPERIMENTS

This section provides the implementation details and discusses the results of our empirical approach.

## 4.1 EXPERIMENTAL SETUP

**Datasets**  The experiments are run on the datasets CIFAR-10 (Krizhevsky, 2009) and Fashion-MNIST (Xiao et al., 2017). These datasets are common classification benchmarks for testing TD in CNNs (Cheng et al., 2021; Denil et al., 2014; Wu et al., 2020; Garipov et al., 2016; Hawkins et al., 2021). For both datasets, the original training sets are split into the set used for training and validation. The split is made such that equal class distributions are maintained. The details specific to the datasets are as follows: **CIFAR-10** has 10 classes distributed equally across 60,000 images of $32 \times 32$ pixels with 3 color channels. After our validation split, there are 45,000 images in the training set and 5,000 in the validation set. The test set of 10,000 images remains unchanged. **Fashion-MNIST** has 10 classes distributed equally across 70,000 grayscale images of $28 \times 28$ pixels. After our validation split, there are 55,000 images in the training set and 5,000 in the validation set.

**Model architecture and training**  The models used are ResNet-18 (He et al., 2016) and GaripovNet (Garipov et al., 2016). ResNet is a well-performing state-of-the-art convolutional neural network. GaripovNet is a 7-layer convolutional neural network proposed in Garipov et al. (2016) and used by Hawkins et al. (2022) for image classification. These models enable comparison with other works within and beyond TD for deep learning (Gusak et al., 2019; Garipov et al., 2016; Hawkins et al., 2022; Chu & Lee, 2021; Kossaifi et al., 2020). The following hyperparameters are used: **ResNet-18** is trained with batch size 128, for 300 epochs, with Adam optimizer and a learning rate of $10^{-3}$. At epochs 100 and 150 the learning rate is multiplied by 0.1. **GaripovNet** is trained with the same settings as the original paper Garipov et al. (2016). The model is trained with Stochastic Gradient Descent (SGD) with a momentum of 0.9 and a learning rate of 0.1, multiplied by 0.1 at epochs 30, 60, and 90.

The validation set is used for early stopping and the selection of the training hyperparameters, i.e. learning rate, schedule, level of annealing, batch size, and optimizer. Training data is augmented with a random crop (padding with 4 pixels and cropping to the original size) and a random horizontal flip. All images are standardized based on the training mean and standard deviation per channel overall training samples. Early stopping is applied for both training the baseline and fine-tuning the decomposed model. The classification error on the test set is used for performance errors $\mathbb{P}$. To Fine-tune after decomposition and obtain performance errors $\mathbb{P}^\star$, the ResNet-18 is optimized for another 25 epochs, and GaripovNet for 10 epochs, using the last learning rate from the training.

**Decomposition choices**  We now explain the considered values for the decomposition choices $\mathbb{H}$ explained in Section 3. For both neural network models, neither the first nor the last layer will be decomposed, as these layers already contain a relatively small amount of parameters. For GaripovNet the other five layers are part of $\mathbb{L}$. For ResNet-18, $\mathbb{L}$ contains a selection of eight convolutional layers, details of which can be found in Appendix A.3. The set of TD methods that will be considered is $\mathbb{M} = \{\text{CP, Tucker, Tensor Train}\}$, which were discussed in Section 3.3. The set of compression levels is $\mathbb{C} = \{10\%, 25\%, 50\%, 75\%, 90\%\}$. Multiple levels of compression are considered as each neural network layer can have different efficiency-performance trade-offs (Lebedev et al., 2015). In the experiments, we evaluate ResNet-18 on CIFAR-10, GaripovNet on CIFAR-10, and GaripovNet on F-MNIST. We exclude ResNet-18 on F-MNIST as the dataset is not sufficiently challenging for this model, and compressing one layer does not lead to a viable impact on the performance due to the model's size and skip-connections.

**Variance**  The process of decomposing and fine-tuning is repeated for five independent runs for each choice of layer, decomposition method, and compression level to assess and report variance in the results. Note that due to the stochasticity of the ALS algorithms, the random initializations can result in different CP decomposition estimates. The variance in correlation shown in the plots without fine-tuning results from the randomness in the CP initialization. Fine-tuning adds additional variance through its use of batched SGD. The observed variance with fine-tuning accounts for both the randomness from CP initialization as well as from fine-tuning, thereby representing all sources of randomness in our methodology. All runs for a given model and dataset are based on the same pretrained weights, so this is not a source of reported variance. In total, this results in 600 measurements for ResNet-18 and 375 measurements for GaripovNet per dataset.

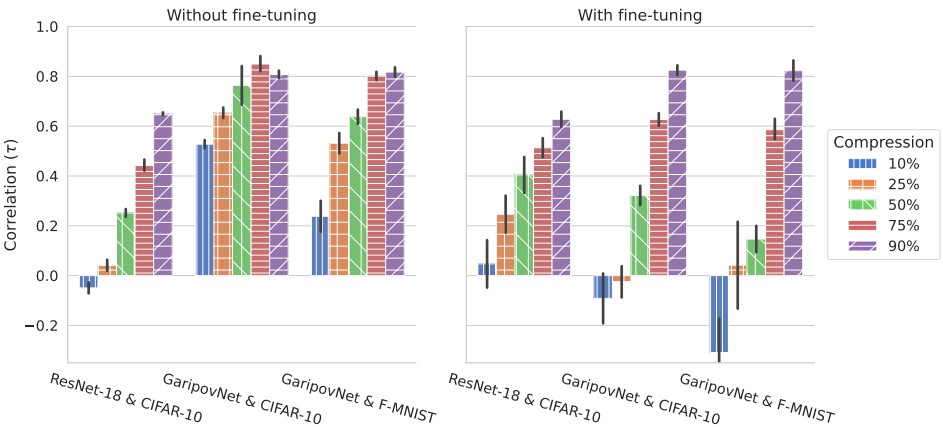

Figure 1: Correlation ($\tau$) over layers and decompositions, grouped by the level of compression (color), using the Relative Weights approximation error. Shown in the bars are averages and standard deviation over runs. We observe that a higher compression results in a larger correlation.

## 4.2 EXPERIMENTAL RESULTS

**Impact of compression levels on correlation**  We start by calculating the correlation across the layers and decomposition methods for multiple runs and calculate the averages grouped by compression levels. This is presented in Figure 1, where the bars are the average correlation $\tau$ between the Relative Weights and the classification error. The correlation is only based on the Relative Weights, as this is the most common metric in recent literature (Lebedev et al., 2015; Hawkins et al., 2022; Liebenwein et al., 2021). The correlations are grouped by the different levels of compression and represented by different colors. The error bars are ±1 standard deviation, representing the variance from multiple runs.

In Figure 1, it can be seen that the larger the compression, the higher the correlation is. This is a positive result for our use case. In the end, we are interested in making decomposition choices when compressing. The more we compress, the higher the correlation and therefore the more certain we are that basing our choice on the approximation error results in the optimal choice. It can also be noted that a certain level of compression is needed to be able to make choices based on the approximation error. For both models and datasets, the correlation is small when the compression is only 10% and 25%. The variance in the correlation at smaller compression levels is larger than at higher compression levels. When the compression is too small the effect on the performance of the model is too small compared to the observed variance, especially after fine-tuning. In the remainder of the experiments, we therefore focus on compression levels of $\mathbb{C} = \{50\%, 75\%, 90\%\}$.

**Comparison of approximation error measures**  Works such as Liebenwein et al. (2021) have used a single approximation error, e.g. Relative Weights, to identify which layer to compress next, implicitly assuming that relative errors between layers are indicative of the relative model performance differences. We here compare the various approximation error measures, testing the correlation with performance over all decomposition choices. In Figure 2, the correlation is calculated based on measurements of all combinations of layer, decomposition method, and compression level once. The correlations are averaged over runs, as well as the ±1 standard deviation is calculated over the runs.

Figure 2 shows that the correlations are generally positive and significantly different from zero. This means that the decomposition choices can (to some extent) be based on the approximation error. There is one exception where the correlation is close to zero, namely for the Absolute Weights measure on ResNet-18. The difference in correlations can be explained by the difference in approximation error between layers, a detailed explanation is provided in Appendix A.4. These results suggest that using Absolute-based approximation errors, while they may show high correlations in some cases, are not generally a reliable indicator for future model performance, and that normalized measures should be used instead.

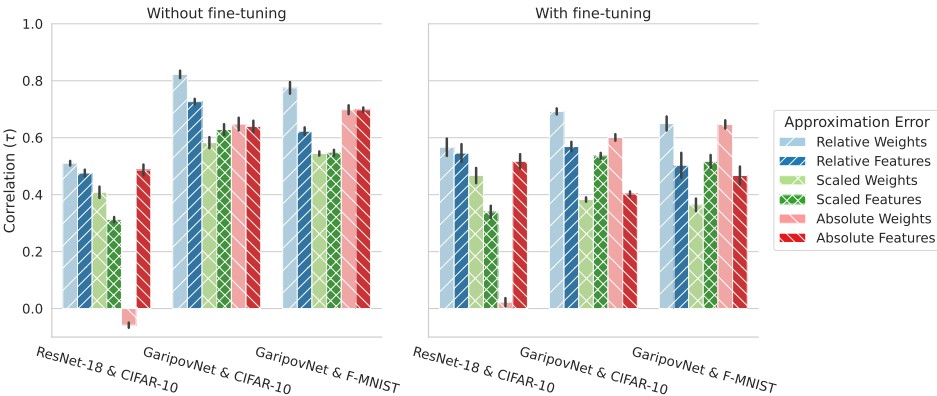

Figure 2: Correlation ($\tau$) calculated over layers, decompositions, and compression levels, averaged (and standard deviation) over runs, for different approximation errors (colors). The Relative Weights approximation error provides the highest correlation, both before and after fine-tuning.

Comparing the different approximation error metrics, we observe the highest correlation with the performance for Relative Weights in all tested cases. Interestingly, the magnitude of the correlation for Feature-based measures is similar to or smaller than the correlation for Weight-based measures. Although the findings of Jaderberg et al. (2014) would suggest a stronger correlation for the features, at least before fine-tuning, we do not observe benefits for basing decomposition choices on the approximation error of the features rather than the weights. Possibly pretraining already ensures all the elements in the weight tensor are equally important for the target data distribution, thus a Weight-based error already reliably reflects resulting errors on the output features. Comparing the error bars with and without fine-tuning, the randomness from fine-tuning has a larger impact on the variance in correlation than the randomness from CP initialization. In summary, our results support the use of the Relative Weights approximation error to make decomposition choices.

**Impact of fine-tuning**  Most works use fine-tuning to recover some of the lost performance (Denton et al., 2014; Lebedev et al., 2015; Kim et al., 2016). The right subfigure of Figure 2 shows the mean and standard deviation of five correlations, per model and per dataset after fine-tuning. After fine-tuning, the correlation between the approximation error and the performance error is smaller than before fine-tuning for GaripovNet, as additional training adapts the model and reduces the performance gap between the different choices, but this effect is not observed for ResNet-18 where the correlation was already lower. However, for both models, there is still a clear positive correlation between the approximation error and the performance after fine-tuning. This means that decomposition choices can still be based on the approximation error when intending to perform fine-tuning later, even though different hyperparameters might be optimal without and with fine-tuning.

While the correlation is positive and significantly different from zero, the correlation is only around +0.5 for ResNet-18. We therefore investigate if the correlation is higher when only considering specific decomposition choices next.

**Correlation across Layers vs. Methods**  In the previous experiments, we compared how decomposition choices on both different layers and methods correlated with performance. Here we investigate if the correlation is stronger if only one of these choices would be considered. For instance, previous works often only include layers as decomposition choice, and have not compared across decomposition methods. We compare correlation on *all* choices for both sets ($\mathbb{L} \times \mathbb{M} \times \mathbb{C}$) as before, to *layers* only ($\mathbb{L} \times \{m\} \times \{c\}$ with reported results averaged for all $m \in \mathbb{M}$ and $c \in \mathbb{C}$), and to *methods* only ($\{l\} \times \mathbb{M} \times \{c\}$ with reported results averaged for all $l \in \mathbb{L}$ and $c \in \mathbb{C}$).

Figure 3 shows that before fine-tuning the approximation error has a lower correlation with the performance of the model when considering layers only compared to all decomposition choices. Not all layers of a neural network have the same efficiency-performance trade-off (Lebedev et al., 2015; Hawkins et al., 2021). Therefore, the correlation is lower when we fix the decomposition method and compression level. It is better to combine layers with compression levels (and decomposition

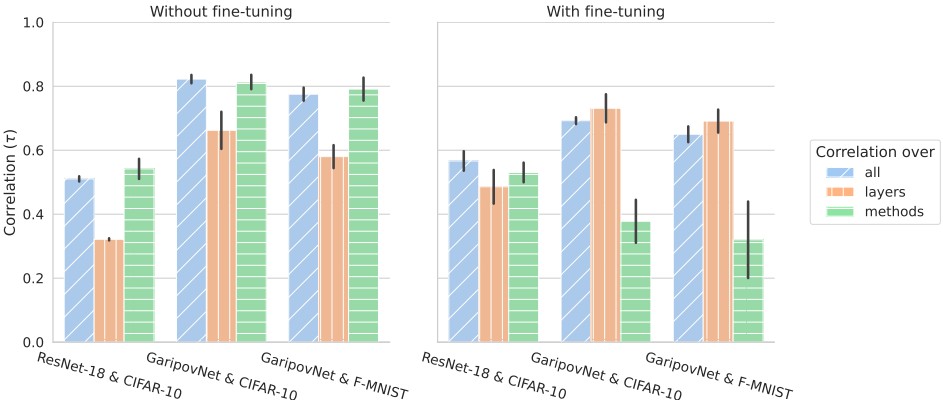

Figure 3: Correlation ($\tau$) calculated over different decomposition choices (colors) for layers and decomposition methods using the Relative Weights error (*all* is identical to its results in Fig. 2). The reported correlations are averaged over all not-compared choices and runs. The standard deviations are only calculated over runs to make the three groups comparable. Before fine-tuning, across layers has less correlation than across methods, though interestingly this pattern is revered by fine-tuning.

methods). However, fine-tuning recovers some of the correlation for layers. Across decomposition methods, the correlation before fine-tuning is comparable to the correlation calculated across all decomposition choices. These results suggest decomposition methods can be compared better than just the layers before fine-tuning, although the former is not an optimization choice considered in previous works. Interestingly, for GaripovNet the correlation across decomposition methods drops significantly after fine-tuning. We find that this is due to difficulties in optimizing the CP decomposed layers, since the gradient flow through CP convolutions is a known problem (Silva & Lim, 2008; Lebedev et al., 2015), whereas ResNet does not suffer from this due to its skip connections. We conclude that (unlike current practice) network compression could consider multiple decomposition methods as their approximation errors can be compared, though most reliably when aiming for compression without fine-tuning.

## 5 CONCLUSION

We have tested Assumption 1, and find that there is a positive correlation between the relative approximation error on the weights and the resulting performance error of the model for a wide range of TD choices, including layers, methods, and compression levels. We further find that using data to compute the approximation error on the features, rather than simply on the model weights directly, does not improve the correlation. Scaling the approximation error with the norm of the original tensor provides the highest and most stable correlation across all compared models and datasets. Our findings suggest that the Relative Weights approximation error is best suited to select among TD decomposition choices.

While these choices can be made across layers, TD methods, and compression levels, we observe that the correlation before fine-tuning is smaller when comparing between layers for a fixed method, than when comparing across methods (here: CP, Tucker, and Tensor Train) for a fixed layer. Integrating multiple types of decompositions within a network compression technique is therefore a potential direction for future work, although care has to be taken when the use case includes later fine-tuning, as the correlation for selecting across decomposition methods can degrade since back-propagation through certain factorized structures remains challenging.

Our experiments are limited to a set of decomposition choices and network layers commonly found in the TD literature. Future work can extend to other decompositions and other types of neural network layers, e.g. fully connected layers. While the weights are matrices, tensor decomposition has been applied to fully connected layers by reshaping the weight matrix into a higher-order tensor. The choice of reshaping then becomes an additional decomposition choice.

## REPRODUCIBILITY

The authors find it important that this work is reproducible. To this extent the following efforts have been made: The datasets and test-validation splits are described in Section 4.1. The datasets are collected from `PyTorch Vision`. The models and hyperparameters used for training are covered in Section 4.1. The implementations of the baseline models are from the `PyTorch Model Zoo`. The models are factorized with `Tensorly-Torch` (Kossaifi et al., 2019b) , using CP initialization described in Section 4.1 and ranks provided in Appendix A.1. The experimental setup is explained in Section 4.1. The calculation of metrics is formulated in Section 3. Finally, the code to reproduce these experiments is available at: `https://github.com/JSchuurmans/tddl`.

## ACKNOWLEDGMENTS

Described results are made possible in part by TU Delft Cohesion subsidy, TERP Cohesion project 2020.

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

# A APPENDIX

## A.1 RANK AND COMPRESSION LEVEL

Tables 2 and 3 present the ranks that are used for GaripovNet (Garipov et al., 2016) and ResNet-18 (He et al., 2016) respectively. Note that in these tables, the Tucker rank includes the kernel ranks corresponding to the width and height, and the TT rank includes $R_0 = R_4 = 1$, which are left out of Equation 4 for conciseness.

Table 2: Ranks of CP, Tucker, and TT corresponding to layers of GaripovNet with a given compression level.

| Layer nr. | Compression (%) | CP | Tucker | TT |
|---|---|---|---|---|
| 2 | 10 | 28 | (14, 14, 3, 3) | (1, 40, 4, 12, 1) |
| 2 | 25 | 69 | (26, 26, 3, 3) | (1, 64, 11, 33, 1) |
| 2 | 50 | 138 | (39, 39, 3, 3) | (1, 64, 27, 64, 1) |
| 2 | 75 | 206 | (49, 49, 3, 3) | (1, 64, 51, 64, 1) |
| 2 | 90 | 248 | (54, 54, 3, 3) | (1, 64, 65, 64, 1) |
| 4 | 10 | 37 | (26, 13, 3, 3) | (1, 63, 6, 18, 1) |
| 4 | 25 | 93 | (49, 24, 3, 3) | (1, 64, 19, 57, 1) |
| 4 | 50 | 186 | (74, 37, 3, 3) | (1, 64, 36, 108, 1) |
| 4 | 75 | 279 | (94, 47, 3, 3) | (1, 64, 60, 128, 1) |
| 4 | 90 | 335 | (105, 52, 3, 3) | (1, 64, 80, 128, 1) |
| 6 | 10 | 56 | (29, 29, 3, 3) | (1, 94, 4, 12, 1) |
| 6 | 25 | 141 | (51, 51, 3, 3) | (1, 128, 21, 63, 1) |
| 6 | 50 | 281 | (77, 77, 3, 3) | (1, 128, 53, 128, 1) |
| 6 | 75 | 422 | (98, 98, 3, 3) | (1, 128, 101, 128, 1) |
| 6 | 90 | 507 | (108, 108, 3, 3) | (1, 128, 130, 128, 1) |
| 8 | 10 | 56 | (29, 29, 3, 3) | (1, 94, 4, 12, 1) |
| 8 | 25 | 141 | (51, 51, 3, 3) | (1, 128, 21, 63, 1) |
| 8 | 50 | 281 | (77, 77, 3, 3) | (1, 128, 53, 128, 1) |
| 8 | 75 | 422 | (98, 98, 3, 3) | (1, 128, 101, 128, 1) |
| 8 | 90 | 507 | (108, 108, 3, 3) | (1, 128, 130, 128, 1) |
| 10 | 10 | 56 | (29, 29, 3, 3) | (1, 94, 4, 12, 1) |
| 10 | 25 | 141 | (51, 51, 3, 3) | (1, 128, 21, 63, 1) |
| 10 | 50 | 281 | (77, 77, 3, 3) | (1, 128, 53, 128, 1) |
| 10 | 75 | 422 | (98, 98, 3, 3) | (1, 128, 101, 128, 1) |
| 10 | 90 | 507 | (108, 108, 3, 3) | (1, 128, 130, 128, 1) |

## A.2 n-MODE PRODUCT

The definition of n-Mode Product $\times_n$ given by Kolda & Bader (2009) is used in this paper. The contraction of a tensor $\mathbf{G} \in \mathbb{R}^{R_1, R_2, \cdots, R_N}$ with matrix $Y \in \mathbb{R}^{Y, R_n}$ along the $n$th mode of the tensor is defined elementwise as:

$$(\mathbf{X} \times_n Y)_{r_1, \cdots, r_{n-1}, y, r_{n+1}, \cdots, r_N} = \sum_{r_n=1}^{R_n} \mathbf{X}_{r_1, \cdots, r_{n-1}, r_n, r_{n+1}, \cdots, r_N} Y_{y, r_n} \tag{5}$$

Table 3: Ranks of CP, Tucker, and TT corresponding to layers of ResNet-18 He et al. (2016) with a given compression level.

| Layer nr. | Compression (%) | CP | Tucker | TT |
|---|---|---|---|---|
| 15 | 10 | 28 | (14, 14, 3, 3) | (1, 40, 4, 12, 1) |
| 15 | 25 | 69 | (26, 26, 3, 3) | (1, 64, 11, 33, 1) |
| 15 | 50 | 138 | (39, 39, 3, 3) | (1, 64, 26, 64, 1) |
| 15 | 75 | 206 | (49, 49, 3, 3) | (1, 64, 51, 64, 1) |
| 15 | 90 | 248 | (54, 54, 3, 3) | (1, 64, 65, 64, 1) |
| 19 | 10 | 37 | (26, 13, 3, 3) | (1, 63, 6, 18, 1) |
| 19 | 25 | 93 | (49, 24, 3, 3) | (1, 64, 19, 57, 1) |
| 19 | 50 | 186 | (74, 37, 3, 3) | (1, 64, 36, 108, 1) |
| 19 | 75 | 279 | (94, 47, 3, 3) | (1, 64, 61, 128, 1) |
| 19 | 90 | 335 | (105, 52, 3, 3) | (1, 64, 80, 128, 1) |
| 28 | 10 | 56 | (29, 29, 3, 3) | (1, 94, 4, 12, 1) |
| 28 | 25 | 141 | (51, 51, 3, 3) | (1, 128, 21, 63, 1) |
| 28 | 50 | 281 | (77, 77, 3, 3) | (1, 128, 53, 128, 1) |
| 28 | 75 | 422 | (98, 98, 3, 3) | (1, 128, 101, 128, 1) |
| 28 | 90 | 507 | (108, 108, 3, 3) | (1, 128, 130, 128, 1) |
| 38 | 10 | 114 | (57, 57, 3, 3) | (1, 207, 5, 15, 1) |
| 38 | 25 | 285 | (103, 103, 3, 3) | (1, 256, 43, 129, 1) |
| 38 | 50 | 569 | (155, 155, 3, 3) | (1, 256, 107, 256, 1) |
| 38 | 75 | 854 | (195, 195, 3, 3) | (1, 256, 203, 256, 1) |
| 38 | 90 | 1025 | (216, 216, 3, 3) | (1, 256, 262, 256, 1) |
| 41 | 10 | 8 | (10, 5, 1, 1) | (1, 256, 11, 33, 1) |
| 41 | 25 | 21 | (25, 12, 1, 1) | (1, 256, 25, 75, 1) |
| 41 | 50 | 42 | (48, 24, 1, 1) | (1, 256, 43, 129, 1) |
| 41 | 75 | 64 | (69, 35, 1, 1) | (1, 256, 213, 256, 1) |
| 41 | 90 | 76 | (82, 41, 1, 1) | (1, 256, 280, 256, 1) |
| 44 | 10 | 114 | (57, 57, 3, 3) | (1, 207, 5, 15, 1) |
| 44 | 25 | 285 | (103, 103, 3, 3) | (1, 256, 43, 129, 1) |
| 44 | 50 | 569 | (155, 155, 3, 3) | (1, 256, 107, 256, 1) |
| 44 | 75 | 854 | (195, 195, 3, 3) | (1, 256, 203, 256, 1) |
| 44 | 90 | 1025 | (216, 216, 3, 3) | (1, 256, 262, 256, 1) |
| 60 | 10 | 229 | (115, 115, 3, 3) | (1, 425, 5, 15, 1) |
| 60 | 25 | 573 | (205, 205, 3, 3) | (1, 512, 85, 255, 1) |
| 60 | 50 | 1145 | (310, 310, 3, 3) | (1, 512, 213, 512, 1) |
| 60 | 75 | 1718 | (390, 390, 3, 3) | (1, 512, 401, 512, 1) |
| 60 | 90 | 2062 | (432, 432, 3, 3) | (1, 512, 526, 512, 1) |
| 63 | 10 | 229 | (115, 115, 3, 3) | (1, 425, 5, 15, 1) |
| 63 | 25 | 573 | (205, 205, 3, 3) | (1, 512, 85, 255, 1) |
| 63 | 50 | 1145 | (310, 310, 3, 3) | (1, 512, 213, 512, 1) |
| 63 | 75 | 1718 | (390, 390, 3, 3) | (1, 512, 401, 512, 1) |
| 63 | 90 | 2062 | (432, 432, 3, 3) | (1, 512, 526, 512, 1) |

## A.3 SELECTION OF RESNET-18 LAYERS

The following layers are considered for decomposition in ResNet-18: The last layer of the first two blocks. The first layer with stride two. The first layer after a 1x1 convolution. A 1x1 convolution with a similar number of parameters as the first choice of layer. The layer before and after the 1x1 convolution. The final two convolutional layers, as they are the largest convolutional layers.

Table 4: The layers considered in ResNet-18. The Layer nr. reflects the layer number in the official PyTorch implementation, starting with the first input layer and counting non-parameterized layers as well.

| Layer nr. | Type of convolution | Order in ResNet block |
|-----------|---------------------|------------------------|
| 15 | regular 2D Conv | last of the first two blocks |
| 19 | regular 2D Conv | first with stride 2 |
| 28 | regular 2D Conv | first after a 1x1 convolution |
| 38 | regular 2D Conv | last layer of two blocks, before 1x1 Conv |
| 41 | 1x1 2D conv | 1x1 conv |
| 44 | regular 2D Conv | first of block, after 1x1 |
| 60 | regular 2D Conv | second to last Conv layer |
| 63 | regular 2D Conv | last Conv layer before avg.pool and classification head |

## A.4 DIFFERENCE BETWEEN LAYERS IN ABSOLUTE WEIGHTS FOR RESNET-18

Let us recall from Figure 2 that when basing the approximation error on Absolute Weights, the correlation with the performance error is close to zero for ResNet-18. The correlation is zero due to the difference between layers.

Figure 4 plots the approximation errors of Relative and Absolute Weights versus the performance error before and after fine-tuning. The points resulting from CP, Tucker, and Tensor Train with compression of 50%, 75%, and 90% are grouped per layer.

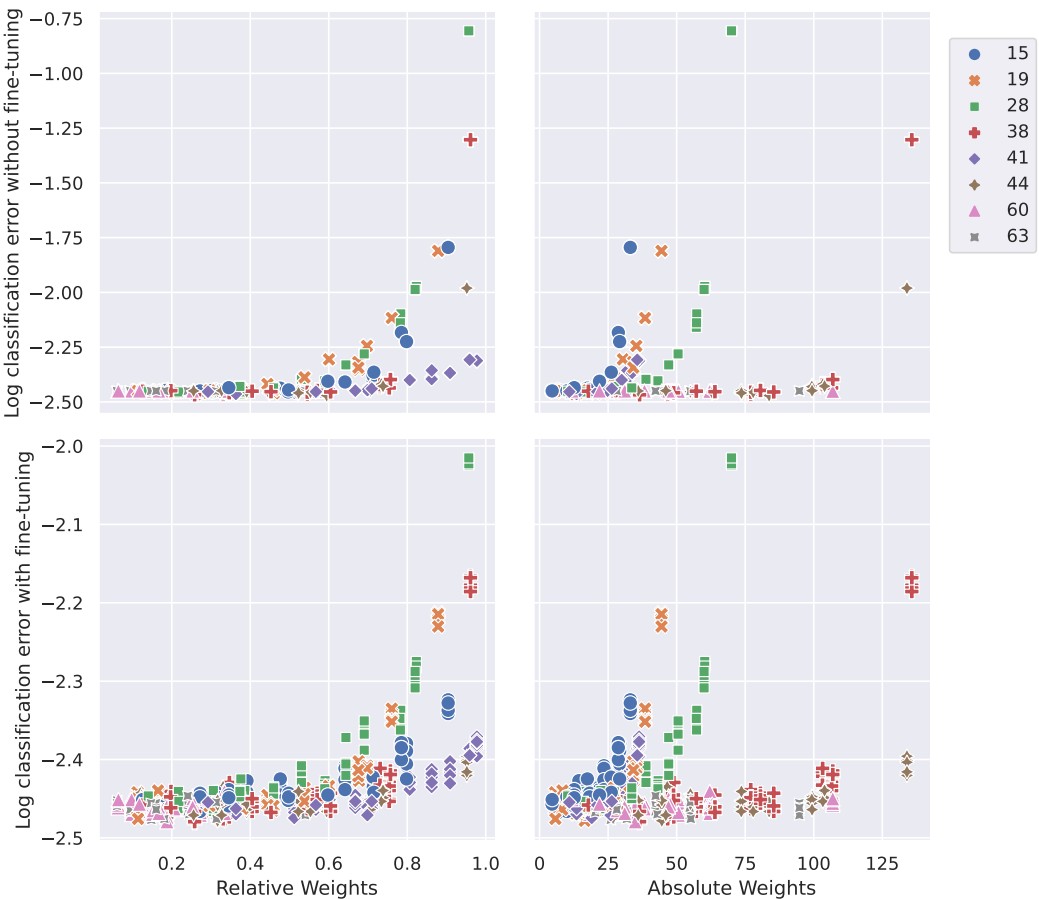

Figure 4: Approximation error calculated with Relative Weights (left) and Absolute Weights (right). Layers 15, 19, and 28 have a smaller absolute weight error and large performance error relative to layers 38, 60, and 63. Compared to the Relative Weights where layers 60 and 63 have a small relative weight error and small performance error and layer 28 has comparable errors to layers 15, 19, and 28.

Layers 15, 19, and 28 have a smaller absolute weight error and large performance error relative to layers 38, 60, and 63. Compare this to the data for Relative Weights where layers 60 and 63 have a small relative weight error and small performance error and layer 28 has comparable errors to 15, 19, and 28. This leads to the correlation between Absolute Weights and performance errors being close to zero, while it is positive for Relative Weights. Therefore, the difference in correlations can be explained by the difference in approximation error between layers.

