# OpenReview forum: "How Informative is the Approximation Error from Tensor Decomposition for Neural Network Compression?"
_ICLR.cc/2023/Conference — ICLR 2023 poster_

### Official Review · Reviewer_ZT8V · 2022-10-22

**Confidence:** 4
**Correctness:** 4
**Technical Novelty And Significance:** 1
**Empirical Novelty And Significance:** 3
**Recommendation:** 8

**Clarity, Quality, Novelty And Reproducibility:**

--- Clarity, Quality ---

The paper is well-written overall, although there are a few places that are unclear. In addition to those points listed under weaknesses above, a few more minor comments follow below.

- The sentence "Across decomposition methods, the correlation before fine-tuning is comparable for all decomposition choices." on page 9 doesn't make sense.
- "Rank" is used both as in tensor decomposition rank, and when discussion the rank correlation throughout the paper. To avoid confusion, I would avoid using "rank" in the second sense, i.e., in reference to the correlation measure.
- In the supplement, I think you should add a definition of the notation $\times_n$ as well as a definition of the Tucker decomposition when the first 2 modes aren't contracted with the core tensor, just so it's clear what "standard" Tucker looks like.
- The colons in the subscripts in Eq (4) are confusing, since this is usually used to denote all indices along a certain mode. It would better if you just explain that $C$ and $T$ are in fact just 2-way tensors here and remove the colons.
- The figures are a bit blurry/pixelated. I would recommend using proper vector graphics.
- The usage of the term "quantization" in Sec 5 is confusing since it's different from how it's typically used in ML; see e.g. https://pytorch.org/docs/stable/quantization.html. I think it would be better to just call it "reshaping" to avoid confusion.

Some minor typos:
- In the 1st sentence of the 2nd paragraph on page 2: The work "make" should be removed. Also "TD decomposition" sounds strange since this reads "tensor decomposition decomposition".
- Last sentence before start of Sec 3.1: TDs -> TD
- There's a period missing in the Fig 1 caption.

--- Novelty ---

The paper doesn't present any novel method, but it does provide a nice empirical investigation of the problem.

--- Reproducibility ---

If the code is released as promised, the paper will be reproducible.

**Strength And Weaknesses:**

--- Strengths ---

S1. The paper provides a good overview of previous works that use TD for compression of NNs.

S2. The problem the paper tries to solve is relevant in practice if someone wants to use TD to compress a CNN, but doesn't have any prior insight into what type of TD would work well for that particular network.

S3. Although some details are hard to understand, the paper is overall well-written and the different experiments are interesting.

--- Weaknesses ---

W1. Some details are hard to understand. I list these below.

a. In Fig 2: What is the difference between Relative Weights and Scaled Weights? Similarly, what is the difference between Relative Features and Scaled Features? Based on Table 1, isn't the relative measures the same as the scaled measures but with the particular choice $n_W = ||W||$ and $n_F = ||F||$? If this is correct, then how is $n_W$ and $n_F$ different for the scaled measures? Also, Table 1 is confusing since $n_W$ and $n_F$ aren't introduced anywhere.

b. In the paragraph "Comparison of approximation error measures" on page 7, in the sentence "In Figure 2, the correlation is calculated ... and rank once.": By "rank", do you mean compression level (i.e., one of the three choices {50%, 75%, 90%})? Calling this rank is a bit confusing, since there's many choices of rank that potentially could yield a certain compression ratio, but only three compression ratios under consideration.

c. Related to the point b above: With 3 types of decomposition and 3 levels of compression and 5 convolutional layers for ResNet-18, shouldn't there be 5 * 3 * 3 = 45 different measurements rather than 40? Similarly, since GaripovNet has 8 layers to choose from, shouldn't there be 8 * 3 * 3 = 72 different measurements for that network rather than 50? Please clarify.

d. In Sec 3, the iterative approach is a bit hard to understand. For example, if my network has 10 layers and I want to compress 5 of them, do I run 5 iterations of the method, compressing one layer each iteration out of the remaining uncompressed layers? Does the iterative method have a target compression, and how does that impact the number of layers to compress? Perhaps you can add a more detailed algorithm for how this works in the supplement?

**Summary Of The Paper:**

The paper considers the use of tensor decomposition (TD) for compression of the weight tensors in CNNs. It considers the problem of choosing the best compression hyperparameters (which layer to compress, which type of TD to use) for a given level of compression. In particular, it investigates if the decomposition error for a given choice can be used as an indicator of how well that particular decomposition choice will perform both without and with fine-tuning when the CNN performs its intended task. If the decomposition error does provide such an indication, it would simplify the task of choosing compression hyperparameters since it would be enough to consider the weight tensor in isolation, rather than how it interacts with the rest of the network.



**Summary Of The Review:**

I like this paper. Although it doesn't introduce any new methods, it provides a nice empirical investigation into the problem of how to choose an appropriate decomposition when using tensor decomposition to compress CNN layers. It is also overall well-written. There are some details that are hard to understand. I listed these four points under weaknesses (a-d under weakness W1). It is particularly important that the reviewers address these four points.

### Update after rebuttal ###

The reviewers addressed the concerns I had. I would have preferred to raise the score from 6 to 7. However, since there is no option for 7, I'm raising it to 8.

---

> ### Author Response · Authors · 2022-11-18
> **Reply to Reviewer ZT8V**
>
> We thank the reviewer for their time and effort and for indicating that this paper provides a good overview, the problem this paper tries to solve is relevant in practice, this paper is overall well-written, and the different experiments are interesting.
>
> ### W1: Hard-to-understand details
>
> Regarding the weakness, we regret to hear some details were hard to understand by the reviewer. We address them below, and improve the paper accordingly:
>
> #### a.
> We previously defined $n_A$ as the number of parameters for any tensor $A$. So tensor $A$ can be the weight tensor $W$ or the feature tensor $F$. Therefore, $n_W \neq || W ||$ and $n_F \neq ||F||$. To make it more clear, we now explicitly define $n_W$ and $n_F$ as the number of elements in the weight tensor and feature tensor respectively in Section 3.2.
>
> #### b.
> You are correct, this should be the compression level. We have changed it accordingly.
>
> #### c.
> Although you switched around ResNet-18 (8 layers) and Garipov (5 layers), you are correct. The calculation was based on an old configuration. We have removed this sentence as we explain the number of measurements elsewhere.
>
> #### d.
> Yes, you understand it correctly. This was a generalization to sketch the context of the problem. It generalizes existing TD compression methods for example those that compress one layer (Jaderberg et al. 2014; Lebedev et al. 2015) and multiple layers (Kim et al., 2016; Goyal et al., 2019; Yin et al., 2022). For the examples we give for one layer, if several layers need to be compressed, then each iteration compresses one layer, as you understood correctly. To achieve a target compression, Kim et al. (2016), Goyal et al. (2019), and Yin et al. (2022) propose different alternatives. Kim et al. (2016) consider a fixed compression level, thereby reducing the number of decomposition choices that need to be made. However, to achieve a target compression, it might be optimal to have different compression levels per layer (Liebenwein et al. 2022; Lebedev et al. 2015; Hawking et al. 2021). Yin et al. (2022) and Goyal et al. (2019) propose an iterative method to compress multiple layers. However, they require extensive evaluation and fine-tuning, which is not needed if the approximation error can be used as an indicator of the performance of the model. We made it more clear in Section 3.
>
> ### Clarity and Reproducibility
> Regarding clarity, _(point 1)_ we changed the sentence.  _(point 2)_ We agree with the reviewer and have changed it accordingly. _(point 3)_ We defined $\times_n$ in Appendix A.2 and added Tucker decomposition when the first 2 modes aren’t contracted in Eq. (3). _(point 4)_ We agree with the reviewer and have changed the formulation accordingly. _(point 5)_ We have changed the figures to a vector format. _(point 6)_ We have changed the usage of quantization (in the TD sense) to reshaping. Our apologies for the typos, they have been addressed.
>
> Regarding reproducibility, we have decided to release the code to the reviewers to improve reproducibility.
>
>
> We hope this clarifies the raised concerns.

---

> > ### Comment · Reviewer_ZT8V · 2022-11-18
> > **Thank you for your response**
> >
> > Thank you for your response and for making those adjustments to the paper. I have raised my score.

---

### Official Review · Reviewer_33s4 · 2022-10-26

**Confidence:** 3
**Correctness:** 3
**Technical Novelty And Significance:** 2
**Empirical Novelty And Significance:** 2
**Recommendation:** 6

**Clarity, Quality, Novelty And Reproducibility:**

The paper is clear, and is reproducible. The major concern is that the contribution is not significant.

**Strength And Weaknesses:**

Strength:
- The paper does a great amount of experiments to validate the assumption, namely there is a positive correlation between the relative approximation error on the weights and the model performance for a wide range of TD choices, including layers, methods, and compression levels.

Weakness:
- The contribution is not significant since the results are not surprised (e.g., relative weight approximation error is better than using absolute weight and rescaled weight), though I appreciate the engineering efforts the author spent on this work.

**Summary Of The Paper:**

The paper performs an empirical study on the relationship between the approximation error of a model’s weight tensor and the model performance after compression. Multiple decomposition methods (CP, Tucker, and Tensor Train), and level of compression are tested on several models and datasets.

**Summary Of The Review:**

As discussed above, I am close to a rejection due to its limited novelty.

---

> ### Author Response · Authors · 2022-11-18
> **Reply to Reviewer 33s4**
>
> We thank the reviewer for their efforts and for recognizing this paper is clear, reproducible, and does a great amount of experiments to validate the assumption. Furthermore, we are encouraged that the reviewer appreciates the engineering efforts spent on this work.
>
> ### W1: The results are not surprising
> We are glad that the reviewer finds our results intuitive to interpret. Regarding the weakness mentioned by the reviewer, we would like to highlight why we believe the results are not self-evident.
>
> - Based on Jaderberg et al. (2014) we did not expect weight-based metrics to outperform feature-based metrics, though our experiments show that it does.
> - Regarding the normalization of the approximation errors mentioned by the reviewer: When comparing compression options across different layers, with different number of parameters and differently sized weights, a case could be made that errors in a large layer with many weights have more impact on network performance, hence large absolute errors by such layers should not be normalized by the number of weights. The alternative case that normalization should be applied is often assumed but has gone untested in the literature, plus there is no clear evidence on how the approximation error should be normalized (with the number of elements in the weight tensor or the norm of the weight tensor). All in all, the outcomes of our experiments relate to how weight errors at different layers affect deep neural network performance in practice, which we believe is not self-evident.
> - We stress that no theoretical derivation is possible to know if assumption 1 holds, as there is no Eckhart-Young theorem (used by Liebenwein et al. (2021) for Matrix decomposition) equivalent for Tensor Decomposition (Vannieuwenhoven et al., 2014).
>
> In conclusion, some of our results may appear straightforward after they have been presented, but knowing for which results this is true and for which not still requires our study.
>
> ### Novelty
> Regarding the novelty, (as recognized by reviewers kjo3 and ZT8V) this paper is the first empirical examination of its kind. Differences with previous work are discussed in the Related Work section. To the best of the authors' knowledge, there is no previous work that studies what this paper studies.
>
> We hope this addresses the raised concerns.

---

### Official Review · Reviewer_YZcC · 2022-11-02

**Confidence:** 3
**Correctness:** 3
**Technical Novelty And Significance:** 2
**Empirical Novelty And Significance:** 3
**Recommendation:** 5

**Clarity, Quality, Novelty And Reproducibility:**

[Clarify score 4/10]
This paper is in general moderately well written. The clarity is not always satisfactory. See Weakness W3.

[Quality score 5/10]

(1) This is the first systematic study on the influences of the approximation error to a compressed neural network’s performance across multiple choices of network layers, TD methods, and compression levels. The experimental findings are new.

(2) Although this work presents extensive experimental results, it lacks deeper explanations for the empirical findings and sufficiently practical suggestions for deep learning practitioners in designing compressed models.

[Novelty score 4/10]

(1) This paper does not propose new models, algorithms, or theory.

(2) It empirically tests the validity of Assumption 1 which is motivated by Liebenwein et al. (2021).

[Reproducibility score 4/10]
The conclusion of this paper relies so heavily on the empirical results. Although the authors gave some detailed descriptions on the experimental settings, there are still so many details not well explained in the current version for reproduction (especially without shared code). For example, what are the detailed model & algorithmic hyperparameters in using CP, Tucker, and TT?

**Strength And Weaknesses:**

[Strengths]

S1: This is the first paper that systematic studies on how well the approximation error relates to a compressed neural network’s performance across multiple choices of network layers, TD methods, and compression levels.

S2: The experimental results support the conclusion that there is a positive correlation between the relative approximation error on the weights and the resulting performance error of the model for a wide range of TD choices, including layers, methods, and compression levels.

[Weaknesses]

W1: To test Assumption 1, this work reports some experimental phenomena/results based on extensive experiments. However, it doesn't go far enough in the sense that possible reasons behind these phenomena (e.g., the impact of compression levels on the rank correlation) are not sufficiently investigated.

W2: Although showing the correlation between weight (tensor) approximation error and the classification performance is interesting, it is more significant if some practical suggestions are given for designing compressed deep models based on tensor decompositions. However, the empirical findings of this work seem of limited guiding value for a deep learning practitioner who needs to decide which layer to compress, which tensor decomposition to use, and which compression level to choose for both efficiency and effectiveness.

W3: The writing is not always satisfactory due to typos like "no works investigate if the make TD decomposition choices using specific approximation errors are well suited", "previous works often only includes", and "Shown are averages and standard deviation over runs We observe that a higher compression". There are also confusing notions used, e.g., "rank" in "the correlation is calculated based on measurements of all combinations of layer, decomposition method, and rank once".



**Summary Of The Paper:**

This paper empirically evaluates the correlation between different hyperparameters of tensor compressive layers and the classification performance. Their results show that the error is indicative of model performance, even when comparing multiple TD methods, though useful correlation only occurs at the higher compression levels.

**Summary Of The Review:**

Due to the evaluation of "Clarity (4/10), Quality (5/10), Novelty (4/10) And Reproducibility (4/10)", I suggest "Reject".

-----------After rebuttal------------
After reading the authors' feedback and other reviewers' comments, I think this is a nice paper which has empirical significance. However, there is still much room for improvement in making this empirical evaluation paper much clearer. So, I decided only to change my suggestion from "3 reject" to "5 borderline below".

---

> ### Author Response · Authors · 2022-11-18
> **Reply to Reviewer YZcC**
>
> We thank the reviewer for the time and effort that went into their review. We are encouraged that the review recognizes this is the first paper that systematically studies how well the approximation error relates to a compressed neural network’s performance, and that the experimental results are new and support the conclusion.
>
> In this reply, we will address the concerns of the reviewer below.
>
> ### W1: Possible reasons behind these phenomena are not sufficiently investigated.
>
> We agree with the reviewer that finding the right explanation is important, though we stress that our study is the first of its kind, hence it is the first to explore and discuss these phenomena for which we test various hypotheses. Our research indeed provides ample leads for further assessment by future work.
>
> Our main hypothesis is that the approximation error is indicative of the performance of the model. We tested this by measuring the correlation between approximation errors and the performance measure. Figure 2 shows this across all TD choices, as it is likely the choices influence each other, and we test if the correlation holds across all choices.
>
> To dive deeper, we investigated the effect of the compression level on the correlation in Figure 1. From this figure, we concluded in Section 4.2 that a larger compression level has a higher rank correlation. We isolated the choice of layers and the choice of decomposition methods in Figure 3. Not all layers of a neural network have the same efficiency-performance trade-off (Lebedev et al., 2015; Hawkins et al., 2021), therefore the correlation is lower when we fix the decomposition method and compression level. It is better to combine layers with compression levels (and decomposition methods). Varying only the decomposition method does not suffer from this effect. This means decomposition methods can be compared better than just the layers, although the opposite is done in previous literature. We have added this deeper explanation to Section 4.2.
>
> ### W2: The empirical findings of this work seem of limited guiding value.
>
> We agree with the reviewer that a practical compression method would be interesting. However, before such an algorithm can be proposed, we argue that is necessary to investigate the underlying assumption (assumption 1) as a theoretical foundation is currently missing. Therefore, the focus of this paper is to lay the empirical foundation for new TD compression methods that exploit the correlations found in this paper, showing under what circumstances the correlation appears, what measures are reliable, and if more choices could be considered than is current practice. In future work, we plan to exploit the correlation found in this work for full model compression.
>
> The guiding value of this paper is therefore the empirical evidence for an assumption where no theory is possible (Vannieuwenhoven et al., 2014). We have made this argument more explicit in Section 1. Furthermore, we provided the motivation for more general decomposition-based compression techniques. Our work shows that it is possible to compare among diverse hyperparameter choices across a heterogenous set of compression techniques.
>
> ### W3: Writing typos
>
> We apologize for the typos and thank the reviewer for pointing out the above-mentioned. They have been fixed and the clarity has been improved.
>
> We hope this addresses the raised concerns.

---

> > ### Comment · Reviewer_YZcC · 2022-11-23
> > **Response to the reply**
> >
> > Thanks for the authors' reply.
> > In the feedback, the authors responded to my concerns about novelty, quality, experimental phenomenon explanation, empirical guiding value, and code sharing.  I agree with the authors that before a practical algorithm can be proposed, it is necessary to investigate whether the underlying assumption (assumption 1) acts as a theoretical foundation. I briefly checked the shared python code, which is complete and clearly written.
> >
> > After reading other reviewers' comments, I think this is a nice paper which has empirical significance. However, there is still much room for improvement in making this empirical evaluation paper much clearer. So, I decided only to change my suggestion from "3 reject" to "5 borderline below".

---

> ### Author Response · Authors · 2022-11-18
> **Clarity, Quality, Novelty And Reproducibility**
>
> # Clarity, Quality, Novelty And Reproducibility
>
> ## Regarding the Quality:
>
> ### (1)
> We thank the reviewer for recognizing this is the first systematic study and that the experimental findings are new.
>
> ### (2)
> We refer the reviewer to our response on W1, where we explain how our experiments give a deeper explanation and how we changed the text to improve the explanation.
>
> ## Regarding the Novelty:
>
> ### (1)
> We refer the reviewer to our response on W2. This motivates why our study is practically relevant and the guiding value of our paper.
>
> ### (2)
> It is important to mention that Liebenwein et al. (2021) use matrix decomposition, for which the Eckhart-Young theorem is used to validate Assumption 1. We use Tensor Decompositions, for which there is no similar theorem possible Vannieuwenhoven et al. (2014) and therefore we need to test it empirically. We have improved the motivation for testing Assumption 1 empirically in Section 1.
>
> ## Regarding reproducibility:
>
> With this response, we are now sharing the (anonymized) code to aid review. As stated in our submission’s Reproducibility statement, our code will be publicly released on GitHub upon acceptance, ensuring all implementation details will be available. Furthermore, we have improved the clarification of the hyperparameters of CP, Tucker, and TT. The estimation of CP decomposition relies on random initialization with a uniform distribution. Tucker and TT have deterministic estimation algorithms mentioned in the paper, HOSVD and TT-SVD respectively. We have added Table 2 and Table 3 in Appendix A1 to clarify which ranks are used for a given compression level.
>
> We hope this addresses the raised concerns.

---

### Official Review · Reviewer_kJo3 · 2022-11-02

**Confidence:** 3
**Correctness:** 4
**Technical Novelty And Significance:** 3
**Empirical Novelty And Significance:** 3
**Recommendation:** 6

**Clarity, Quality, Novelty And Reproducibility:**

The paper was clear, succint. The experiments are reproducible and all their components, down to the exact correlation that were used have been clearly described. Apart from the two weakness mentioned I have no other concerns.

**Strength And Weaknesses:**

S1. The paper investigates a clear objective that is relevant for potential follow-up work and has useful practical implications.

S2. The paper is well written and well structured.

S3. The paper does give a clear picture and considers many practically relevant questions.


W1. It is unclear how the increased complexity due to the proposed methods and the level of control on the final task performance do compare against a distillation approach.

W2. How much does the randomness of many of the TD methods due to their initialization influence to robustness of the approach.

**Summary Of The Paper:**

In this study the authors investigate post-training compression of CNN-based image classification neural networks. One method to compress a CNN-based NN is tensor decomposition. Their goal is to learn how much the approximation error of tensor decomposition is predictive of the performance of the compressed CNN-based NN classifier. For example, they ask if the choice of tensor decomposition method can be guided by this or the level of compression. This would be more efficient than using the final task performance of each candidate compressed model. Another question, for example, they investigate is if finetuning after compression does disturb or improve the correlation between approximation error and final task performance of the NN.

**Summary Of The Review:**

The paper is well written, focused and has enough experiments to educate the reader about their findings.

---

> ### Author Response · Authors · 2022-11-18
> **Reply to Reviewer kJo3**
>
> We thank the reviewer for their time and efforts, and for recognizing this paper investigates a clear objective that is relevant, is well-written, gives a clear picture, and considers many practically relevant questions.
>
> In this reply, we will address the concerns of the reviewer below.
>
> ### W1: How does the proposed approach compare to distillation?
>
> We agree with the reviewer that it will be necessary to compare a TD-based compression method to distillation and other compression techniques, although we believe this should be the next step. Our paper provides the key insights needed to motivate and develop general TD-based compression techniques. Due to the extensiveness of our empirical study, turning our insights into a complete compression method is out of scope, and left as future work.
>
> ### W2: Influence of randomness to robustness?
>
> This is an excellent observation, and we will clarify the effect of randomness on our results better in Sections 4.1 and 4.2. In short, TT and Tucker decomposition are estimated deterministically and only CP relies on randomness in the initialization. The effect of randomness from CP initialization is illustrated by the error bars in figures without fine-tuning (Figures 1-3, left). The results with fine-tuning (Figures 1-3, right) include both the variation resulting from the randomness of CP initialization, as well as randomness due to fine-tuning with batched SGD. Therefore, our analysis does account for the randomness of the TD methods. The error bars show that the variance due to randomness is relatively small compared to the compared correlations, except for the lower compression levels in Figure 1 when the rank correlations are also limited.
>
> We hope this addresses the raised concerns.

---

### Author Response · Authors · 2022-11-18
**Rebuttal Revision & Code**

The changes due to the reviewers' comments are in red and the code is provided in the Supplementary Material.

---

### Decision · Program_Chairs · 2023-01-20

**Decision:**

Accept: poster

**Justification For Why Not Higher Score:**

This is a nice paper which has empirical significance. However, there is still much room for improvement in making this empirical evaluation paper much clearer.  This paper does not propose new models, algorithms, or theory.

**Justification For Why Not Lower Score:**

The rebuttal do have resolved several major concerns raised by reviewers.  The impressive empirical significance of this work is well received.

**Metareview: Summary, Strengths And Weaknesses:**

Summary: The paper performs an empirical study on the relationship between the approximation error of a model’s weight tensor and the model performance after compression. Multiple decomposition methods (CP, Tucker, and Tensor Train), and level of compression are tested on several models and datasets. Although it doesn't introduce any new methods, it provides a nice empirical investigation into the problem of how to choose an appropriate decomposition when using tensor decomposition to compress CNN layers. It is also overall well-written.

Strengths:  The paper investigates a clear objective that is relevant for potential follow-up work and has useful practical implications. This is the first paper that systematic studies on how well the approximation error relates to a compressed neural network’s performance across multiple choices of network layers, TD methods, and compression levels.

Weakness: Although this work presents extensive experimental results, it lacks deeper explanations for the empirical findings and sufficiently practical suggestions for deep learning practitioners in designing compressed models.

**Note From Pc:**

if the above contains the word "oral" or "spotlight" please see: "oral" presentation means -> notable-top-5% and "spotlight" means -> notable-top-25%. As stated in our emails, we are disassociating presentation type from AC recommendations